# Genetic variants determine intrafamilial variability of SARS-CoV-2 clinical outcomes in 19 Italian families

Alessia Azzarà[1]☯*, Ilaria Cassano[1,2]☯, Elisa Paccagnella[1], Maria Cristina Tirindelli[3], Carolina Nobile[3], Valentina Schittone[4], Carla Lintas[1,5], Roberto Sacco[1,6], Fiorella Gurrieri[1,5]

**1** Research Unit of Medical Genetics, Department of Medicine and Surgery, Università Campus Bio-Medico di Roma, Rome, Italy, **2** Dipartimento di Scienze della Vita e di Sanità Pubblica, Sezione di Medicina Genomica, Università Cattolica del Sacro Cuore, Rome, Italia, **3** Transfusion Medicine and Cellular Therapy Unit, Fondazione Policlinico Universitario Campus Bio-Medico, Rome, Italy, **4** Hematology and Stem Cell Transplant Unit, Fondazione Policlinico Universitario Campus Bio-Medico, Rome, Italy, **5** Operative Research Unit of Medical Genetics, Fondazione Policlinico Universitario Campus Bio-Medico, Rome, Italy, **6** Operative Research Unit of Neurodevelopmental Disorders/Child Neuropsychiatry, Fondazione Policlinico Universitario Campus Bio-Medico, Rome, Italy

☯ These authors contributed equally to this work.
* a.azzara@unicampus.it

**Data Availability Statement:** All relevant data are within the paper and its Supporting Information files.

## Abstract

Severe acute respiratory syndrome coronavirus 2 (SARS-CoV-2) infection results in a wide range of outcomes characterized by a high heterogeneity in both symptomatology and susceptibility to the disease. In such a perspective, COVID-19 may be considered as a multifactorial disease featured by the interaction between the environment, which is the virus itself, and the genetic profile of the host. Our analysis aimed at investigating the transmission dynamics of SARS-CoV-2 within families whose members responded in different ways to the infection, although the exposure was common to the entire group and occurred before the availability of any vaccine. The goal was to understand how the genetic background of each subject can influence the viral infection outcome and hence the above-mentioned clinical variability. We performed a segregation analysis in 19 Italian families with a designed custom panel of 42 genes involved in immunity and virus entry and which have also been shown to be related to SARS-CoV-2 host response. We carried out both a familial segregation analysis and a global statistical analysis. In the former we identified eighteen risk variants co-segregating with a COVID-positive status and six variants with a possible protective effect. In addition, sixteen variants showed a trend of association to a severe phenotype. Together with common SNPs, we detected private rare variants that may also provide insight into the observed clinical COVID-19 heterogeneity. The global statistical analysis confirmed statistically significant positive associations between SARS-CoV-2 individual response and some specific gene variants identified in familial analysis. In conclusion our data confirm that the clinical expression of COVID-19 is markedly influenced by the host genetic profile both with a mendelian transmission pattern and a polygenic architecture.

**Funding:** AA, CL, IC received a fellowship for one year by Rome Biomedical Campus University Foundation (AA), the ANIA Associazione Nazionale fra le Imprese Assicuratric Foundation (CL) and Fundraising area of the Rome Biomedical Campus University (IC). https://www.unicampus.it/ateneo/biomedical-university-foundation https://www.ania.it/ https://sostienici.unicampus.it/ The funders had no role in study design, data collection and analysis, decision to publish, or preparation of the manuscript.

**Competing interests:** The authors have declared that no competing interests exist.

## Introduction

Severe Acute Respiratory Syndrome Coronavirus 2 (SARS-CoV-2) is the etiological agent responsible for Coronavirus disease 2019 (COVID-19), firstly reported in China in December 2019. Since then, the virus has rapidly spread worldwide causing millions of deaths to date. The number of deaths is still high (above 300 victims per day in Italy), in spite of the actual wide and effective vaccination campaign.

COVID-19 clinical picture is characterized by a high heterogeneity ranging from asymptomatic or mild symptoms in the majority of infected people (~ 80%) to moderate-severe symptoms in ~20% hospitalized individuals of which about 5% die. Mild common symptoms include dry cough, fever, asthenia, anosmia and ageusia. On the other hand, severe and often fatal clinical manifestations in hospitalized individuals include interstitial pneumonia with hypoxemia, acute respiratory distress syndrome, sepsis and multiorgan failure. Moderate and severe symptoms have been positively correlated with patient age, male gender and comorbidities.

With respect to SARS-CoV-2 pathogenesis, it is known that it all starts with the virus entry into the host cell through the high affinity binding of the spike (S) glycoprotein to the human angiotensin-converting enzyme 2 (ACE2). Indeed, this first viral-host interaction elicits an overactivation of the angiotensin II type receptor axis, which in turn causes strong vasoconstriction and at the same time activates profibrotic, proapoptotic and proinflammatory signal cascades [1]. Because of such a dysregulation, the lungs and the cardiovascular system are among the most damaged organs with an increased risk of thromboembolic events in severely affected hospitalized patients [2].

The heterogeneity in disease symptomatology and susceptibility is mediated by the complex interaction between the virus and the host genetic make-up: COVID-19 can be considered a multifactorial disease in which the main environmental component is the SARS-CoV-2 itself, whereas the genetic component is the human host genetic profile. Indeed, many research efforts have pointed to understand how the genetic background of each individual can influence the viral infection outcome, considering the reported extreme clinical variability in disease symptomatology and susceptibility [3]. Different genome wide association studies (GWAS) have found significant signals in chromosomal regions where genes involved in immune function (3p21.31) and in the ABO blood group system (9q34.2) localize [4]. A recent GWAS performed on hospitalized and non-hospitalized patients in United Arab Emirates identified additional significant peaks in eight new loci harboring genes expressed in the lungs and associated to emphysema, airway obstruction, lung surface tension and other genes associated with innate and adaptive immune responses [5]. Furthermore, case-control studies have also found significant associations of specific single nucleotide polymorphisms (SNPs) with COVID-19 severity and susceptibility. The list of relevant genes includes the tumor necrosis factors TNFα and TNFß [6], interleukin 6 [7], hemostatic genes (*PROC*, *MTHFR*, *MTR*, *ADAMTS13* and *THBS2*) [8], the *ACE* gene [9] and the *ACE2* gene [10]. In addition to common SNPs, rare variants may also contribute to the observed clinical COVID-19 heterogeneity. Indeed, several exome sequencing studies have been performed with the aim of identifying common and rare variants contributing to COVID-19 outcome. *Latini et al.* [11] observed significant differences in the frequency of some rare and common variants relative to *TMPRSS2* and *PCSK3* genes in COVID-19 patients compared to the GnomAD control population whereas no difference was reported for genetic variants relative to the *ACE2* gene [12]. Recent work by *Picchiotti et al.* [13] proposes a model named "post-Mendelian genetic model" in which the number of both common and rare host-specific gene variants contribute to a polygenic score, able to predict the severity of COVID-19 for each individual.

In spite of a wealth of genetic studies, to date no systematic analysis has focused on the transmission dynamics of SARS-CoV-2 phenotypes and genotypes within families, even though clinical variability in symptomatology and infection-susceptibility is regularly observed within the same multiplex family in which siblings share a consistent genetic load. With the aim of explaining such unexpected intrafamilial variability, we performed a segregation analysis of variants in COVID-related genes in multiplex families with variable clinical manifestations (from asymptomatic to severe). We applied targeted NGS sequencing to 22 Italian families of four to ten members each with high internal clinical variability. We designed a customized panel of 42 genes, which have been shown to be related to COVID-19 host response, also adding genes involved in immunity and virus entry.

## Materials and methods

### Patients

We enrolled 19 families with members infected or not with COVID-19 and with different severity of disease (Fig 1). We considered the families with four or more co-habiting members that have a positive/negative SARS-CoV-2 test and/or a positive serological test and/or typical symptoms after exposure to a confirmed COVID-19 case. We clustered COVID-19 positive patients in three groups: asymptomatic, mild and severe, this latter according to clinical features like pneumonia, hypoxemia, dyspnoea and/or high fever, or hospital admission. For all of them, we also reported the presence of anosmia/ageusia (Fig 1 and Table 1). Each family member was interviewed with respect to the symptomatology of COVID-19: "severity" phenotype was attributed to those who had respiratory issues, desaturation or pneumonia; moderately affected those who had fever, pain, cough or tiredness; asymptomatic were those with confirmed laboratory test without clinical features.

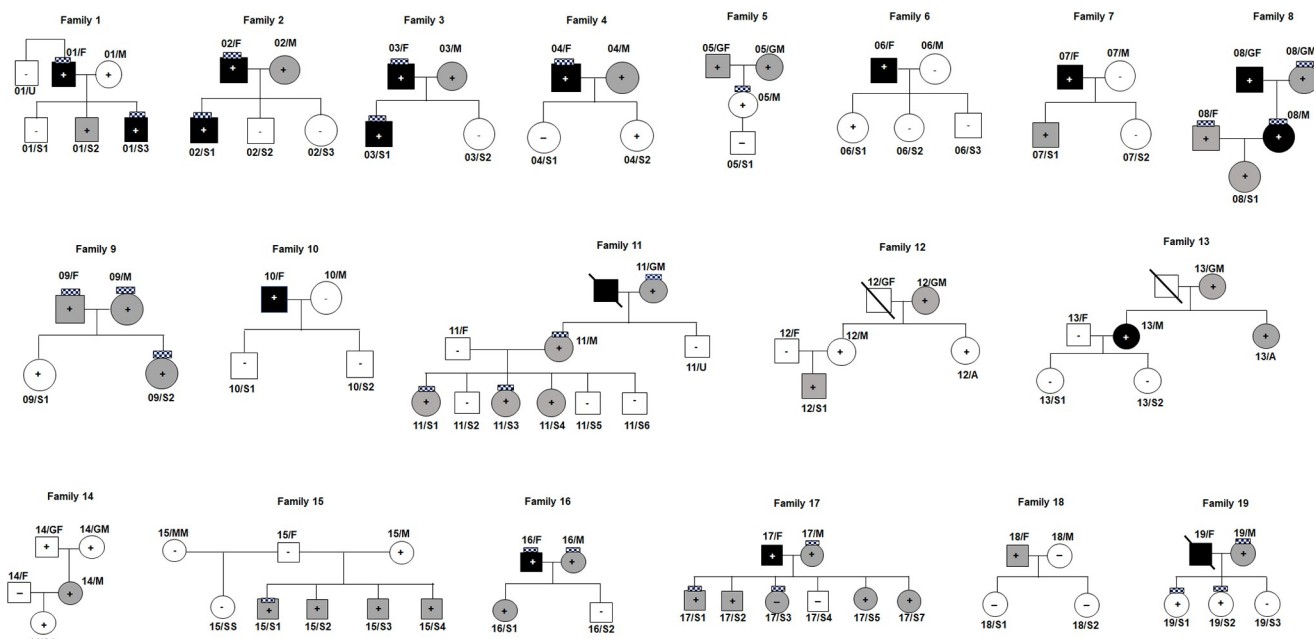

**Fig 1. Pedigrees of the eligible families.** White squares and circles with the "minus" sign indicate SARS-CoV-2 negative subjects. White squares and circles with the "plus" sign indicate SARS-CoV-2 positive and asymptomatic subjects. Grey squares and circles with the "plus" sign indicate SARS-CoV-2 positive subjects with a mild phenotype. Dark squares and circles with the "plus" sign indicate SARS-CoV-2 positive subjects with severe symptoms. The "chess" symbol stands for subjects with anosmia/ageusia.

**Table 1. Clinical features of our families.**

| Family N° | Patient ID | Gender | Age | SARS-CoV-2 test | Symptoms | Anosmia | Hospitalization | Comorbidities | Additional information |
|---|---|---|---|---|---|---|---|---|---|
| 1 | 01/F | Male | 53 | + | Severe | Yes | No | No | |
| | 01/M | Female | 46 | + | Asymptomatic | Yes | No | No | |
| | 01/S1 | Male | 18 | - | / | / | / | / | |
| | 01/S2 | Male | 16 | + | Mild | No | No | No | |
| | 01/S3 | Male | 13 | + | Severe | Yes | No | No | |
| | 01/G | Male | 17 | - | / | / | / | / | |
| | 01/U | Male | 59 | - | / | / | / | / | |
| 2 | 02/F | Male | 60 | + | Severe | Yes | No | No | |
| | 02/M | Female | 51 | + | Mild | No | No | No | |
| | 02/S1 | Female | 20 | + | Severe | Yes | No | No | |
| | 02/S2 | Female | 18 | - | / | / | / | / | |
| | 02/S3 | Male | 15 | - | / | / | / | / | |
| 3 | 03/F | Male | 60 | + | Severe | Yes | No | No | |
| | 03/M | Female | Unknown | + | Mild | No | No | No | |
| | 03/S1 | Male | 20 | + | Severe | Yes | No | No | |
| | 03/S2 | Female | 15 | - | / | / | / | / | |
| 4 | 04/F | Male | 51 | + | Severe | Yes | No | No | Dyspnoea; high fever |
| | 04/M | Female | 50 | + | Mild | No | No | No | Headache |
| | 04/S1 | Female | 15 | - | / | / | / | / | |
| | 04/S2 | Female | 12 | + | Mild | No | No | No | |
| 5 | 05/GF | Male | Unknown | + | Mild | No | No | No | Muscle pain; cough |
| | 05/GM | Female | Unknown | + | Mild | No | No | No | Muscle pain; asthenia |
| | 05/F | Male | 49 | + | Mild | No | No | No | |
| | 05/M | Female | Unknown | + | Asymptomatic | Yes | No | No | |
| | 05/S1 | Male | 9 | - | / | / | / | / | |
| 6 | 06/F | Male | Unknown | + | Severe | No | No | No | Hypoxemia; high fever; muscle pain; asthenia |
| | 06/M | Female | Unknown | - | / | / | / | / | |
| | 06/S1 | Female | 10 | + | Asymptomatic | No | No | No | |
| | 06/S2 | Female | 12 | - | / | / | / | / | |
| | 06/S3 | Male | 8 | - | / | / | / | / | |
| 7 | 07/F | Male | 55 | + | Severe | No | Yes | No | Bilateral pneumonia |
| | 07/M | Female | 52 | - | / | / | / | / | |
| | 07/S1 | Male | 18 | + | Mild | No | No | No | |
| | 07/S2 | Female | 23 | - | / | / | / | / | |
| 8 | 08/GF | Male | 59 | + | Severe | No | No | No | Pneumonia; cough |
| | 08/GM | Female | 55 | + | Mild | Yes | No | No | Diarrhea; asthenia |
| | 08/F | Male | 39 | + | Mild | Yes | No | No | Chest pain |
| | 08/M | Female | 34 | + | Severe | Yes | No | No | Pneumonia |
| | 08/S1 | Female | 6 | + | Mild | No | No | No | |
| 9 | 09/F | Male | 45 | + | Mild | Yes | No | No | Muscle pain |
| | 09/M | Female | 43 | + | Mild | Yes | No | No | Fever for 2–3 days |
| | 09/S1 | Female | 11 | + | Asymptomatic | No | No | No | |
| | 09/S2 | Female | 14 | + | Mild | Yes | No | No | Headache |

(*Continued*)

**Table 1.** (Continued)

| Family N° | Patient ID | Gender | Age | SARS-CoV-2 test | Symptoms | Anosmia | Hospitalization | Comorbidities | Additional information |
|---|---|---|---|---|---|---|---|---|---|
| 10 | 10/F | Male | 54 | + | Severe | No | Yes | No | Hypoxemia; fever; muscle pain |
| | 10/M | Female | 50 | - | / | / | / | / | |
| | 10/S1 | Male | 21 | - | / | / | / | / | |
| | 10/S2 | Male | 14 | - | / | / | / | / | |
| 11 | 11/GM | Female | 77 | + | Mild | Yes | No | No | Her husband died of CoViD-19; dyspnoea; muscle pain; asthenia |
| | 11/U | Male | 53 | - | / | / | / | / | |
| | 11/F | Male | 54 | - | / | / | / | / | |
| | 11/M | Female | 53 | + | Mild | Yes | No | No | Fever; headache; cough; muscle pain; asthenia |
| | 11/S1 | Female | 27 | + | Mild | Yes | No | No | Muscle pain; cold |
| | 11/S2 | Male | 26 | - | / | / | / | / | |
| | 11/S3 | Female | 23 | + | Mild | Yes | No | No | Muscle pain; cold |
| | 11/S4 | Female | 21 | + | Mild | / | No | No | Fever; muscle pain; cold |
| | 11/S5 | Male | 20 | - | / | / | / | / | |
| | 11/S6 | Male | 16 | - | / | / | / | / | |
| 12 | 12/GM | Female | 69 | + | Mild | No | No | No | Dyspnoea; muscle pain; asthenia |
| | 12/A | Female | 48 | + | Asymptomatic | No | No | No | |
| | 12/F | Male | 49 | - | / | / | / | / | |
| | 12/M | Female | 46 | + | Asymptomatic | No | No | No | |
| | 12/S1 | Male | 15 | + | Mild | No | No | No | Fever |
| 13 | 13/GM | Female | 83 | + | Mild | No | No | Alzheimer disease | |
| | 13/A | Female | 59 | + | Mild | No | No | No | |
| | 13/F | Male | Unknown | - | / | / | / | / | |
| | 13/M | Female | 49 | + | Severe | No | No | No | |
| | 13/S1 | Female | 24 | - | / | / | / | / | |
| | 13/S2 | Female | 22 | - | / | / | / | / | |
| 14 | 14/GF | Male | 71 | + | Asymptomatic | No | No | Diabetes; heart disease | |
| | 14/GM | Female | 65 | + | Asymptomatic | No | No | Diabetes; hypertension | |
| | 14/F | Male | 44 | - | / | / | / | / | |
| | 14/M | Female | 41 | + | Mild | No | No | No | Sore throat; diarrhea |
| | 14/S1 | Female | 7 | + | Asymptomatic | No | No | No | |
| 15 | 15/MM | Female | 49 | - | / | / | / | / | |
| | 15/SS | Female | 19 | - | / | / | / | / | |
| | 15/F | Male | 53 | - | / | / | / | / | |
| | 15/M | Female | 54 | + | Asymptomatic | No | No | No | |
| | 15/S1 | Male | 28 | + | Mild | Yes | No | No | Fever; headache |
| | 15/S2 | Male | 25 | + | Mild | No | No | No | Fever for 2–3 days; cough |
| | 15/S3 | Male | 22 | + | Mild | No | No | No | Fever for 2–3 days; headache |
| | 15/S4 | Male | 26 | + | Mild | No | No | No | Fever for 2–3 days; asthenia; cough |
| 16 | 16/F | Male | 55 | + | Severe | Yes | No | No | Dyspnoea; high fever; nausea |
| | 16/M | Female | 55 | + | Mild | Yes | No | No | Low-grade fever; headache; muscle pain |
| | 16/S1 | Female | 22 | + | Mild | No | No | No | Low-grade fever |
| | 16/S2 | Male | 24 | - | / | / | / | / | |

*(Continued)*

**Table 1.** (Continued)

| Family N° | Patient ID | Gender | Age | SARS-CoV-2 test | Symptoms | Anosmia | Hospitalization | Comorbidities | Additional information |
|---|---|---|---|---|---|---|---|---|---|
| 17 | 17/F | Male | 54 | + | Severe | No | No | No | Hypoxemia; fever; cough |
| | 17/M | Female | 55 | + | Mild | Yes | No | No | Fever; muscle pain |
| | 17/S1 | Male | 25 | + | Mild | Yes | No | No | High fever for 2 days |
| | 17/S2 | Male | 24 | + | Mild | No | No | No | |
| | 17/S3 | Female | 20 | + | Mild | Yes | No | No | Fever |
| | 17/S4 | Male | 18 | - | / | / | / | / | |
| | 17/S5 | Female | 15 | + | Mild | No | No | No | Fever |
| | 17/S7 | Female | 11 | + | Mild | No | No | No | High fever for 3–4 days |
| 18 | 18/F | Male | 46 | + | Mild | No | No | No | Low-grade fever; diarrhea; muscle pain; cough for 1 month |
| | 18/M | Female | 45 | - | / | / | / | / | |
| | 18/S1 | Female | 10 | - | / | / | / | / | |
| | 18/S2 | Female | 7 | - | / | / | / | / | |
| 19 | 19/F | Male | 53 | + | Severe | No | Yes | Diabetes | Died of CoViD-19 after 7 days of fever and hypoxemia |
| | 19/M | Female | 52 | + | Mild | Yes | No | No | Muscle pain |
| | 19/S1 | Female | 23 | + | Asymptomatic | Yes | No | No | |
| | 19/S2 | Female | 20 | + | Asymptomatic | Yes | No | No | |
| | 19/S3 | Female | 12 | - | / | / | / | / | |

+ or -: Positive or negative SARS-CoV-2 test result.

/: Absent.

We included in the analysis 19 families for an overall number of 102 patients (55 female subjects and 47 male ones), of whom 35 were SARS-CoV-2 negative and 67 were SARS-CoV-2 positive. In particular, 12 out of 67 were asymptomatic patients, 39 out of 67 had a mild phenotype of the disease and 16 out of 67 had severe symptoms. Finally, 28 out of 67 had anosmia/ageusia. Patients' age ranged from 6 to 83 years, with a mean age of 34,7 years (SD = 19,5), and 54% of the patients were female.

The "COVID-19 positive" status or "COVID-19 negative" status represented a screenshot of the clinical situation during the exposure to the original SARS-CoV-2 spread and in the temporal window from October 2020 to December 2020, before the availability of any vaccine. Families were enrolled through a collaboration with the Haematological Transfusion Medicine Unit that was involved in recruitment of hyperimmune post COVID individuals. An informed written consent was collected, including a specific one for minors signed by parents. The local Institutional Review Board of Campus Bio-Medico University of Rome approved this study (IRB n° 04.21). Genomic DNA was extracted from saliva by our in-house protocol.

## Next-generation sequencing

We designed a custom panel covering coding exons and splice junctions of 42 genes involved in autoimmune and autoinflammatory pathways and in response to viral infections; some of these genes were reported in association to SARS-CoV-2 infection in the host genome studies [14, 15] (S1 Table). Our Ampliseq custom panel covered 99,9% of the targeted regions. The remaining 0,01% mapped in a little region of 5'UTR that we did not consider. The panel contained 734 amplicons. The custom panel was manufactured by Illumina.

Briefly, libraries were prepared using the Ampliseq manifacture and then pooled and loaded into MiSeqDx for sequencing. Sequencing data were analysed using the Illumina Local Run Manager. Annotation was performed by uploading VCF files into wAnnovar. Only variants covered with more than 20 reads were filtered in and were manually checked on BAM file using the UCSC Genome Browser. The synonymous substitutions that did not affect splicing were filtered out. The remaining other variants were described based on their frequency in the gnomAD database, also indicating the rs number of dbSNP.

For the variants with a MAF lower than 0.01 in gnomAD, we reported the evaluation of *in silico* tools by Varsome [16] and the CADD (combined annotation-dependent depletion) score [17]. In the UCSC Genome Browser, there is a track set of the GWAS Data Release 4 (October 2020) from the COVID-19 Host Genetics Initiative (HGI) [18, 19], that shows the results of meta-analysis across multiple studies by partners world-wide to identify the genetic determinants of SARS-CoV-2 infection susceptibility, disease severity and outcomes. For this reason, we reported whether the variant has been found in the HGI meta-analyses data A2, B2, C1, and C2. The four tracks in UCSC Genome B describe the data as: Severe COVID-19 variants (A2) like cases with very severe respiratory failure confirmed for COVID-19 vs. population (i.e. everybody that is not a case); Hospitalized COVID-19 variants (B2) like cases hospitalized and confirmed for COVID-19 vs. population (i.e. everybody that is not a case); tested COVID-19 variants (C1) like cases with laboratory confirmed SARS-CoV-2 infection, or health record/physician-confirmed COVID-19, or self-reported COVID-19 via questionnaire vs. laboratory /self-reported negative cases. Finally, all COVID-19 variants (C2) like cases with laboratory confirmed SARS-CoV-2 infection, or health record/physician-confirmed COVID-19, or self-reported COVID-19 vs. population (i.e. everybody that is not a case). The tracks are colored in red for risk effect or in blue for protective effect and the color saturation indicates effect size (beta coefficient): values over the median of effect size are darkly colored, those below the median are colored less intensively.

We performed both familial segregation analysis and global statistical analysis.

## Familial analysis

In the familial analysis, we evaluated segregating variants in every single family unit, in relation to: 1) the presence of variants only in the family members with a positive SARS-CoV-2 test (and absent in the negative ones); 2) the presence of variants only in the family members negative to the SARS-CoV-2 test (and absent in the positive ones); 3) variants shared in the positive SARS-CoV-2 family members based on the three groups of symptomatology (asymptomatic, mild and severe phenotype). Building on this analysis, our aim was to confirm previous or identify new variants or genes related to: 1) risk of infection, 2) protection from infection, and possibly likelihood of manifesting severe symptoms of COVID-19.

## Global statistical analysis

In spite of the small size of our sample, we performed a statistical analysis to check whether the variants showed association with either COVID-negative or COVID-positive test and verify the overall correlation of each genetic variant with the severity of symptoms and the presence of anosmia. Clinical parameters and genetic markers were analysed using a categorical approach using either $\chi^2$, T1 $\chi^2$ statistics or Fisher Test. T1 $\chi^2$ statistics were performed using the CLUMP software, whenever contingency tables contained more than 25% of cells with expected values <5. The odds ratios for estimating the risk were also calculated. One-way ANOVAs was applied to analyse the relationship between age and SARS-CoV-2 test. Global analysis was performed on 102 subjects (67 COVID-positive / 35 COVID-negative). We

considered the following variables in the analysis: gender, age, COVID-positive / COVID-negative status, presence of symptoms and level of severity, presence of anosmia and gene mutation rate (*i.e.* how much that gene was mutated) in COVID-positive / COVID-negative subjects. Data are reported as frequencies (%) or mean ± S.D, depending on the type of variable; statistical significance is set at a nominal two-tail P<0.05. Statistical analyses were performed using STATA software release 14.0, IBM SPSS software release 24.0; the EPI INFO 7 software was used to calculate the odds ratio (OR).

## Results

We performed a custom panel sequencing of 42 genes in 19 families segregating COVID-19 disease. These genes are involved in autoimmune and autoinflammatory pathways, in cell entry and in viral infection response, or have been reported in host genomic studies. We performed two types of analysis: A) a familial analysis and B) a global statistical analysis.

### Familial analysis (variants co-segregating with phenotype)

In the familial analysis, we reported the variants that segregate with a COVID-positive status, a COVID-negative status and a symptomatic status in individual families (Tables 2–4). We excluded conflicting variants (i.e. variants present in members of different families independently of the clinical status). Complete lists of these variants in this study are available in S2–S5 Tables. Segregation patterns of the identified variants within families are shown in Fig 2.

**Risk variants.** In Table 2, we listed the 18 risk variants segregating with the COVID-positive status in 11 of 19 families.

**Table 2. Familial segregation of risk variants in families with positive SARS-CoV-2 test.**

| Position [a] | Ref | Alt | Gene | AA change | GnomAD MAF % [b] | dbSNP | COVID-19 Host Genetics Initiative | Segregating families |
|---|---|---|---|---|---|---|---|---|
| chr2:163128824 | T | C | IFIH1 | p.His843Arg | 67,6 | rs3747517 | | 8, 9 |
| chr3:46010077 | C | T | FYCO1 | p.Arg250Gln | 84,6 | rs4683158 | | 8, 9, 10 |
| chr3:46009864 | C | G | FYCO1 | p.Gly321Ala | 56,5 | rs3733100 | | 8, 10, 14 |
| chr3:46007823 | GTT | TTC | FYCO1 | p.Asn1001Glu | 12 | rs13079478, rs13059238 | A2,B2,C2 | 10, 14 |
| chr3:46009487 | G | A | FYCO1 | p.Arg447Cys | 12,1 | rs33910087 | A2,B2,C2 | 14 |
| chr3:45837886 | G | C | SLC6A20 | p.Ala9Gly | 11,7 | rs2271615 | C1, C2 | 12 |
| chr2:113537223 | C | A | IL1A | p.Ala114Ser | 26,8 | rs17561 | | 12 |
| chrX:12903659 | A | T | TLR7 | p.Gln11Leu | 17,9 | rs179008 | | 12 |
| chr3:46008820 | C | T | FYCO1 | p.Ser669Asn | 0,15 | rs141155944 | | 10 |
| chr3:46008841 | G | A | FYCO1 | p.Ser662Phe | 0,15 | rs150785981 | | 10 |
| chr19:7831628 | G | A | CLEC4M | p.Asp291Asn | 26,5 | rs2277998 | | 1 |
| chr20:3838441 | C | G | MAVS | p.Gln93Glu | 28 | rs17857295 | | 4 |
| chr4:187004074 | C | T | TLR3 | p.Leu412Phe | 27,5 | rs3775291 | | 5 |
| chr1:247587343 | G | A | NLRP3 | p.Val200Met | 0,83 | rs121908147 | | 6 |
| chrX:12924826 | A | G | TLR8 | p.Met1? | 30,5 | rs3764880 | | 16 |
| chr14:103342049 | T | C | TRAF3 | p.Met129Thr | 32,8 | rs1131877 | | 18 |
| chr20:3843027 | C | A | MAVS | p.Gln198Lys | 14,5 | rs7262903 | | 8 |
| chr2:163124051 | C | T | IFIH1 | p.Ala946Thr | 49,7 | rs1990760 | | 9 |

[a] Human GRCh37/hg19

[b] Minor Allele Frequency %

Red: variants already reported as of risk by the COVID-19 HGI in the UCSC Genome Browser.

**Table 3. Familial segregation of protective variants in families with negative SARS-CoV-2 test.**

| Position [a] | Ref | Alt | Gene | AA change | gnomAD MAF % [b] | dbSNP | COVID-19 Host Genetics Initiative | Segregating families |
|---|---|---|---|---|---|---|---|---|
| chr21:42852497 | C | T | TMPRSS2 | p.Val197Met | 24,9 | rs12329760 | B2 | 5, 10 |
| chr11:614318 | T | C | IRF7 | p.Lys192Glu | 26,1 | rs1061502 | | 10, 18, 15 |
| chr21:34715699 | G | C | IFNAR1 | p.Val168Leu | 18 | rs2257167 | | 10 |
| chr11:320805 | G | T | IFITM3 | p.His3Gln | 4,1 | rs1136853 | | 10 |
| chr21:42866297 | G | A | TMPRSS2 | p.Thr112Ile | 0,7 | rs61735793 | | 18 |
| chr17:5485367 | A | T | NLRP1 | p.Leu155His | 36,2 | rs12150220 | | 6 |

[a] Human GRCh37/hg19

[b] Minor Allele Frequency %

Blue: variants already reported as of protection by the COVID-19 HGI in the UCSC Genome Browser

Three families (n°8, n°9, n°12) carried several variants in different genes. Some of these variants (total four out of eighteen) were in common between different families: the p.His843Arg in *IFIH1* gene, three variants in *FYCO1* gene (p.Arg250Gln, p.Gly321Ala and p.Asn1001Glu) and the last one substitution was also identified as a risk variant by the HGI study.

The other fourteen variants were non-shared between families and present in single ones. Among them, we detected two risk variants co-segregating with COVID-positive status in families n°14 and n°12 (the p.Arg447Cys in *FYCO1* gene and the p.Ala9Gly in *SLC6A20* gene, respectively) previously reported in the HGI study. In family n°12, also the p.Gln11Leu in *TLR7* and p.Ala114Ser in *IL1A* co-segregated with COVID-19 positive status.

Two other variants in family n°10 (p.Ser669Asn and p.Ser662Phe in *FYCO1* gene) were exclusively detected in this family in *cis*, both with a frequency of 0,15% in gnomAD.

**Table 4. Familial segregation of variants in families with severe COVID-19.**

| Position [a] | Ref | Alt | Gene | AA change | gnomAD MAF % [b] | dbSNP | COVID-19 Host Genetics Initiative | Segregating families |
|---|---|---|---|---|---|---|---|---|
| chr19:7831628 | G | A | CLEC4M | p.Asp291Asn | 26,5 | rs2277998 | | 6,7 |
| chr11:613978 | C | T | IRF7 | p.Gly260Arg | 0,069 | rs201379782 | | 6 |
| chr11:614799 | C | T | IRF7 | p.Arg144Gln | 0,06 | rs201036875 | | 6 |
| chr3:45801393 | T | C | SLC6A20 | p.Ile529Val | 0,72 | rs61731475 | | 6 |
| chr9:32500832 | C | T | DDX58 | p.Arg71His | 0,9 | rs72710678 | | 6 |
| chr9:120475302 | A | G | TLR4 | p.Asp299Gly | 6,1 | rs4986790 | | 6 |
| chr9:120475602 | C | T | TLR4 | p.Thr399Ile | 5,6 | rs4986791 | | 6 |
| chr11:102248377 | C | T | BIRC2 | p.Ala506Val | 4,8 | rs34510872 | | 6 |
| chr3:45869972 | C | T | LZTFL1 | p.Asp246Asn | 7,4 | rs1129183 | | 6 |
| chrX:12903659 | A | T | TLR7 | p.Gln11Leu | 17,9 | rs179008 | | 8 |
| chrX:12924826 | A | G | TLR8 | p.Met1? | 30,5 | rs3764880 | | 8 |
| chr3:45837886 | G | C | SLC6A20 | p.Ala9Gly | 11,7 | rs2271615 | C1, C2 | 13 |
| chr3:45987980 | G | A | CXCR6 | p.Glu3Lys | 4,4 | rs2234355 | | 13 |
| chr3:46007846 | C | T | FYCO1 | p.Glu994Lys | 2,6 | rs34801630 | | 13 |
| chr9:32480251 | A | T | DDX58 | p.Asp580Glu | 10 | rs17217280 | | 16 |
| chr4:103534740 | - | A | NFKB1 | c.2749+11dupA | 6,5 | rs148268461 | | 19 |

[a] Human GRCh37/hg19

[b] Minor Allele Frequency %

Red: variants already reported as of risk by the COVID-19 HGI in the UCSC Genome Browser.

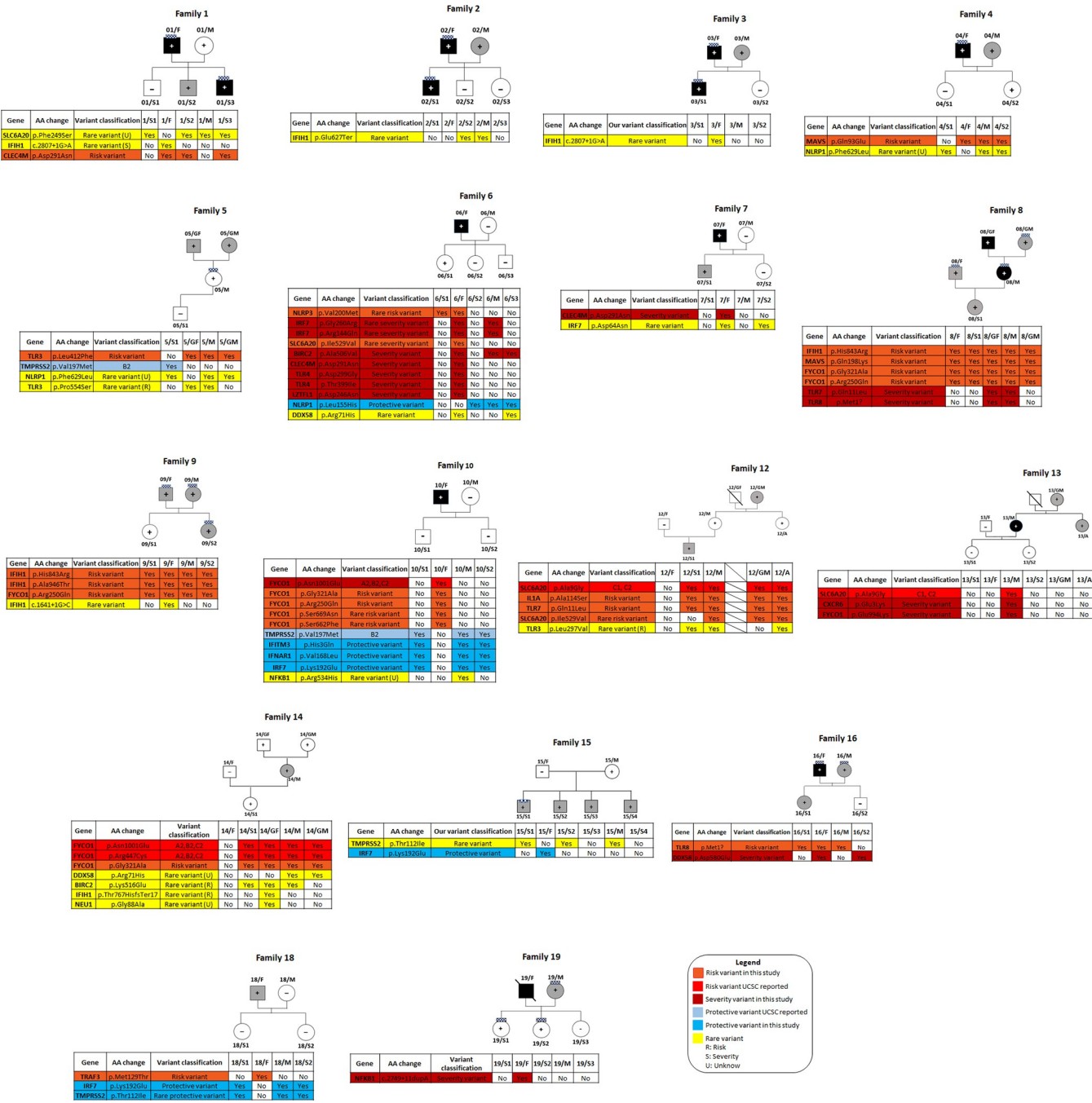

**Fig 2. Significant genetic variants with respect to COVID-19 susceptibility and protection.** Red: UCSC risk variants; light blue: UCSC protective variants; orange: our risk variants; blue: our protective variants; bordeaux: our severity variants; yellow: unknown correlation between rare variants and phenotype.

In six families (n°1, n°4, n°5, n°6, n°16, n°18), single risk variants segregated: the p. Asp291Asn in *CLEC4M* gene (MAF 26.5%), the p.Gln93Glu in *MAVS* gene (MAF 28%), the p. Leu412Phe in *TLR3* gene (MAF 27,5), the p.Val200Met in *NLRP3* gene (MAF 0,83%), the p. Met1? in *TLR8* gene (MAF 30,5%), the p.Met129Thr in *TRAF3* gene (MAF 32,8%), respectively.

**Protective variants.** In Table 3, we reported six variants present only in COVID-negative members in a single family and/or were shared by other families with a possible protective effect.

Out of six protective variants, one was shared between families n˚5 and n˚10 (p.Val197Met in *TMPRSS2* gene). This variant showed co-segregation with COVID-negative status and it was previously reported as protective by the HGI GWAS study. Another protective variant (p. Lys192Glu in *IRF7* gene) was shared between families n˚10, n˚18 and n˚15. Families n˚10 and n˚18 showed additional protective variants. In particular, in family n˚10 we detected the p. Val168Leu in *IFNAR1* gene and the p.His3Gln in *IFITM3* gene. In family n˚18 the p.Thr112Ile variant in *TMPRSS2* gene was carried by COVID-negative members (MAF 0.7%). Family n˚6 segregated only a single non-shared variant in *NLRP1* gene.

**Clinical severity variants.** Performing the 3) analysis of variants association to the severity of symptoms, we did not find any variant segregating with the asymptomatic status or a mild phenotype in the families, but we detected sixteen variants in patients with severe COVID-19 (Table 4). Among them, the only variant shared by two families (n˚6 and n˚7) was the p.Asp291Asn in *CLEC4M* gene. The same variant segregates with COVID-positive status also in family n˚1, where two members were severely affected and one moderately.

Out of the other fifteen variants, eight were detected in family n˚6, four of which are rare (two in the *IRF7* gene, one in *SLC6A20* and one in *DDX58*). Three variants, two in family n˚8 and one in family n˚13 were already considered as risk variants in other families (p.Gln11Leu in *TLR7*, p.Met1? in *TLR8*, p.Leu412Phe in *TLR3*, p.Ala9Gly in *SLC6A20* gene) and they were found to be associated with severe phenotype although two COVID-negative individuals in family n˚2 were also carriers. In addition, we found two other variants (in *CXCR6* and in *FYCO1*) in family n˚13.

In families n˚16 and n˚19 we found the variant p.Asp580Glu in *DDX58* gene (MAF 10%) and the variant c.2749+11dupA in *NKFB1* gene (MAF 6.5%) in association with severe disease.

**Rare variants.** In our study, we have detected forty-three variants that have a MAF <1% or that are absent in the gnomAD database (S4 Table). These variants were found in single/ multiple members of the families, but did not always co-segregate in members sharing the same symptomatology or the same COVID-test result. In Table 5, we have reported only twenty variants: twelve rare variants evaluated as "damaging" and eight rare variants that were mentioned in previous tables for their compatible segregation with a risk/protective model or with a severe phenotype in the families.

Among these, three rare risk variants segregate with a SARS-CoV-2 positive test in the families. Only the variant p.Ser662Phe in *FYCO1* gene (family n˚10) has a high CADD score (24), with an evaluation of a possible "damaging" effect. The other two variants have a low CADD score and were scored as "tolerated". The only rare variant segregating in family n˚18 with a "protective" pattern (the p.Thr112Ile in the *TMPRSS2* gene; prediction "tolerated"), was present in four positive members of family n˚15, three mildly affected and one asymptomatic. Lastly, the four rare variants (two in the *IRF7* gene, one in *SLC6A20* gene and one in *DDX58* gene) segregating in a patient with a severe phenotype of COVID-19 (family n˚6), were all evaluated as "tolerated" with a CADD score <25. The two variants in *IRF7* gene are also present in his unrelated COVID-negative wife. The variant in *SLC6A20* gene is present in three positive members of family n˚12 (two asymptomatic cases and one mild). The substitution in *DDX58* gene is reported in two COVID-positive patients of family n˚14, one asymptomatic and another with mild phenotype.

The twelve remaining variants with a CADD score >25 and/or globally evaluated as "damaging" by VarSome tools did not follow a specific segregation of risk/protective pattern. Among them, the variant p.Lys516Glu in *BIRC2* gene, the c.1641+1G>C and p.

**Table 5. Rare variants (MAF<1%) identified in subjects of our families.**

| Position [a] | Ref | Alt | Gene | AA change | gnomAD MAF % [b] | dbSNP | CADD score | VarSome tools in silico prediction | N° Positive | | | N° Negative | Clinical classification | Patient ID |
|---|---|---|---|---|---|---|---|---|---|---|---|---|---|---|
| | | | | | | | | | Severe | Mild | Asintomatic | | | |
| chr1:247587343 | G | A | NLRP3 | p.Val200Met | 0,83 | rs121908147 | 0 | 9/10 tolerated | 1 | | 1 | | R | 6F; 6S1 |
| chr3:46008820 | C | T | FYCO1 | p.Ser669Asn | 0,15 | rs141155944 | 14 | 11/12 tolerated | 1 | | | | R | 10F |
| chr3:46008841 | G | A | FYCO1 | p.Ser662Phe | 0,15 | rs150785981 | 24 | 7/12 damaging | 1 | | | | R | 10F |
| chr21:42866297 | G | A | TMPRSS2 | p.Thr112Ile | 0,7 | rs61735793 | 9 | 11/11 tolerated | | 3 | 1 | 3 | U | 15M;15S1; 15S2; 15S4; 18M;18S1; 18S2 |
| chr11:613978 | C | T | IRF7 | p.Gly260Arg | 0,069 | rs201379782 | 0 | 11/12 tolerated | 1 | | | 1 | S | 6F;6M |
| chr11:614799 | C | T | IRF7 | p.Arg144Gln | 0,06 | rs201036875 | 12 | 11/12 tolerated | 1 | | | 1 | S | 6F;6M |
| chr3:45801393 | T | C | SLC6A20 | p.Ile529Val | 0,72 | rs61731475 | 18 | 9/11 tolerated | 1 | 1 | 2 | | R | 6F; 12A; 12M; 12GM |
| chr9:32500832 | C | T | DDX58 | p.Arg71His | 0,9 | rs72710678 | 23 | 11/17 tolerated | 1 | 1 | 1 | 1 | U | 6F;6S3; 14M; 14GM |
| chr11:102248406 | A | G | BIRC2 | p.Lys516Glu | 0,15 | rs61754131 | 27 | 8/12 damaging | | 1 | 2 | | R | 14M; 14S1; 14GF |
| chr2:163136505 | C | G | IFIH1 | c.164I+1G>C | 0,67 | rs35337543 | 32 | 6/6 damaging | | 1 | | | R | 9F |
| chr2:163133202 | T | - | IFIH1 | p.Thr767HisfsTer17 | 0,0057 | rs759430873 | 34 | 1/1 damaging | | | 1 | | R | 14GF |
| chr4:187004500 | C | T | TLR3 | p.Pro554Ser | 0,04 | rs121434431 | 23 | 9/12 damaging | | 1 | 1 | | R | 5M; 5GF |
| chr4:187003729 | C | G | TLR3 | p.Leu297Val | 0,15 | rs35311343 | 23 | 7/11 damaging | | 1 | 2 | | R | 12A; 12M; 12S1 |
| chr2:163124596 | C | T | IFIH1 | c.2807+1G>A | 0,6 | rs35732034 | 33 | 6/6 damaging | 2 | | | 1 | S | 1F; 1U; 3F |
| chr11:615129 | C | T | IRF7 | p.Asp64Asn | - | - | 22 | 10/12 damaging | 1 | | 1 | 1 | S | 7F; 7S2 |
| chr6:31829865 | C | G | NEU1 | p.Gly88Ala | 0,75 | rs34712643 | 27 | 8/10 damaging | | | 1 | | U | 14GF |
| chr4:103518782 | G | A | NFKB1 | p.Arg534His | 0,053 | rs150281816 | 26 | 8/12 damaging | | | | 1 | U | 10M |
| chr17:5462129 | G | T | NLRP1 | p.Phe629Leu | 0,15 | rs149035689 | 26 | 10/17 damaging | | 3 | 1 | 2 | U | 4M; 4S1; 4S2; 5M; 5GM; 5S1; |
| chr3:45812898 | A | G | SLC6A20 | p.Phe249Ser | 0,1 | rs147760034 | 28 | 11/12 damaging | 1 | 1 | 1 | 1 | U | 1M; 1S1; 1S2;1S3 |
| chr2:163134090 | C | A | IFIH1 | p.Glu627Ter | 0,31 | rs35744605 | 38 | 5/5 damaging | | 1 | | 1 | U | 2M; 2S2 |

R: Risk variant; S: Severity variant; U: Unclassified; /: Absent

[a] Human GRCh37/hg19

[b] Minor Allele Frequency

Thr767HisfsTer17 in *IFIH1* gene, the p.Pro554Ser and the p.Leu297Val in *TLR3* gene were detected in single/multiple subjects with COVID-positive test result. The variant p.Pro554Ser in *TLR3* gene was previously reported as risk variant [15, 20]. In addition, the splicing variant c.2807+1G>A in the *IFIH1* gene (CADD score 33) was present in two patients from families n˚1 and n˚3 with a severe phenotype. This variant might affect the splicing event with a mechanism of donor loss by *in silico* predictions (SpliceAI score 0,980), likely representing a variant associated with a severe symptomatology. The variant p.Asp64Asn in *IRF7* gene (absent in gnomAD database), present in one positive patient with a severe phenotype and in one negative case, may similarly act as the potentially severe variant described above. The last five damaging variants were distributed equally between COVID-positive/negative subjects. **Global statistical analysis**

Global statistical analysis in our population showed that the age is significantly higher in subjects with COVID-19 positive test compared to negative (average age positive = 38,9 *vs.* average age negative = 26,6; ANOVA: F = 8,550, 1 df, p = 0,004). Among positive individuals, the symptomatology is significantly associated with sex (asymptomatic females = 91,7% *vs.* asymptomatic males = 8,3%; chi square = 6.978, p = 0.008).

With respect to variants, results of the global statistical analysis are summarized in Table 6, reporting the statistical value (*i.e.* odd ratio, confidence interval, chi-squared test, p-value). In this table, we presented only the variants and genes that showed a significant association in our population in relation to the different variables taken into account (see Material and Methods in global analysis section).

We found significant association (Table 6) between variant p.Ala9Gly in *SLC6A20* gene and p.Asp291Asn variant in *CLEC4M* gene with COVID-19 positive status (positive individuals = 50% *vs.* negative = 29.4%; positive individual s = 57.6% *vs.* negative = 31.4%, respectively), as well as significant association of the variant p.Ala506Val in *BIRC2* gene with more severe symptoms of COVID-19 (Asymptomatic = 12.5%; Mild symptoms = 25%; Severe symptoms = 62.5%).

**Table 6. Statistical global analysis of the gene variants.**

|  | OR (95% CI) | Chi square test | df | P-value |
|---|---|---|---|---|
| **RISK VARIANTS** |  |  |  |  |
| *SLC6A20* p.Ala9Gly | 2.40 (0.97–5.93) | 3.67 | 1 | 0.05/0.04[a] |
| *CLEC4M* p.Asp291Asn | 1.85 (0.80–4.25) | 7.30 | 1 | 0.03 |
| **SEVERITY VARIANTS** |  |  |  |  |
| *BIRC2* p.Ala506Val | 1.79 (0.19–16.32) | 8.39 | 2 | 0.015 |
| **VARIANTS BY GENDER DIFFERENCE** |  |  |  |  |
| *BIRC2* p.Ala506Val | 6.18 (1.12–34.14) | 5.20 | 1 | 0.023 |
| *MAVS* p.Ala218Cys | 0.27 (0.09–0.77) | 6.20 | 1 | 0.013 |
| **MOST MUTATED GENE IN POSITIVE INDIVIDUALS** |  |  |  |  |
| *STAT2* | 1.57 (1.51–1.62) | 4.55 | 1 | 0.03 |
| **MOST MUTATED GENE IN NEGATIVE INDIVIDUALS** |  |  |  |  |
| *IFITM3* | 0.14 (0.015–1.26) | 7.06 | 1 | 0.007 |
| **VARIANTS ASSOCIATED WITH ANOSMIA** |  |  |  |  |
| *IFIH1* p.Ala946Thr | 0.21 (0.03–1.21) | 7.60 | 2 | 0.022 |
| *MAVS* p.Arg218Cys | 0.72 (0.26–1.93) | 6.60 | 2 | 0.037 |
| *TLR8* p.Met1 | 5.00 (1.51–16.46) | 7.46 | 2 | 0.006 |
| *TRAF3* p.Met129Thr | 2.99 (1.03–8.65) | 4.21 | 2 | 0.04 |

[a] Refers to the fisher test applied only in this case

In addition, with respect to gender difference among COVID-positive subjects, this variant in *BIRC2* gene is more represented in positive males (positive females = 5.7% *vs.* positive males = 27.3%), whereas the variant p.Ala218Cys in *MAVS* gene is more represented in positive females (positive females = 63.2% *vs.* positive males = 32.1%).

Finally, the most frequently mutated gene in positive individuals was *STAT2* (positive subjects = 0.62% *vs.* negative = 0%), whereas the gene most frequently mutated in negative individuals was *IFITM3* (positive subjects = 0% *vs.* negative = 0.55%).

We also detected several variants significantly associated with anosmia: the p.Ala946Thr in *IFIH1* gene (presence of anosmia = 36.4% *vs.* absence = 17.6%), the p.Arg218Cys in *MAVS* gene (presence of anosmia = 22.2% *vs.* absence = 5.3%), the p.Met1? in *TLR8* gene (presence of anosmia = 71.4% *vs.* absence = 33.3%) and the p.Met129Thr in *TRAF3* gene (presence of anosmia = 48.1% *vs.* absence = 23.7%).

## Discussion

Our study focused on families whose members reacted differently to SARS- COV2 infection in spite of an identical and contemporary exposure to the agent within the domestic environment.

The approach of familial segregation analysis of candidate genes variants has the advantage to allow precise and punctual attribution of the correct clinical status in a restricted time window in which the exposure to infection was common to the entire family. Previous genomic data (either GWAS or whole exome analysis) on large cohorts of unrelated affected and unaffected SARS-COV 2 subjects have made clear that genomic profiles can make the difference in the individual response to COVID-19 infection. GWAS studies have identified common variants in relevant genes [4, 21] and the Host Genomic Initiative as well as the Italian GEN-COVID Multicenter Study have in addition identified both rare and common candidate variants that influence host response to COVID-19 infection. Many of these variants are currently listed in UCSC genome browser as protective or risk variants. We included these genes in our panel and added new genes because of their functional involvement in cellular response to viral infection.

With respect to risk variants, we confirmed the UCSC report for different variants in *FYCO1*, segregating with COVID-19 positive status in several families and also identified four new risk variants in the same gene. Five of the above variants segregate as a disease haplotype in individual 10F, who is severely affected. One other variant, p.Glu994Lys is associated with a severe phenotype in individual 13M, although it does not co-segregate with COVID-19 positive status in other affected individuals in this family. We hypothesize that some variants cause a severe clinical status only in presence of other risk variants with a permissive effect but, in absence of those variants, they may not be sufficient to even determine COVID-positive status. *FYCO1* maps within a cluster on chromosome 3 reported in association with severe SARS-COV-2 in a large GWAS study [4]. Altogether, our data confirm that *FYCO1* is most likely a susceptibility gene and that the more variants are carried by the host, the higher is the susceptibility to viral infection. Loss of function variants in *FYCO1* cause autosomal recessive congenital cataract likely due to a defect in autophagy of damaged lens structures [22]. *FYCO1* is involved in autophagy and in autophagosome/lysosome fusion. Although all the *FYCO1* variants detected in our COVID-19 positive individuals are missense and likely benign variants, we hypothesize that such variants might modulate the intracellular clearance in damaged tissues and clog the cell, worsening tissue damage.

Within the same risk-gene cluster in chromosome 3, we also detected two *SLC6A20* variants, segregating with COVID-19 positive status. The involvement of *SLC6A20* is meaningful,

as its product functionally interacts with angiotensin-converting enzyme 2, the SARS-CoV-2 cell-surface receptor. SLC6A20 can form heterodimers with ACE2 and BOAT1, able to function as binding sites for SARS-CoV-2 spike glycoproteins likely impacting the ability of ACE2 to mediate viral infection [23, 24]. Global statistical analysis showed significant association of *SLC6A20* variants with COVID positive status, confirming its role in disease susceptibility.

We detected potential new risk variants in three members of the TLR gene-family: *TLR3*, *TLR7* and *TLR8*. We found two variants in *TLR4* gene, already reported in a study of 300 Egyptian patients with a significantly positive risk of severe COVID-19 [25]. Variant in *TLR3* gene, was previously reported as a marker of severity [13, 26]. However, the variant was transmitted to two COVID negative sons. It is known that Toll-like receptors (TLR) play an important role in recognition of viral particles and activation of the innate immune system leading to the secretion of pro-inflammatory cytokines, such as interleukin-1 (IL-1), IL-6, and tumor necrosis factor-α, as well as type 1 interferon. The latter has been recently reported as a crucial biomarker of severe SARS-CoV-2 disease [21]. In addition, TLRs might represent potential target in controlling the infection in the early stages of disease [27].

One new additional risk variant was found in the *CLEC4M* gene, segregating with severe COVID-19 infection. CLEC4M has been reported as an alternative binding receptor for SARS-CoV-2 spike glycoprotein, likely mediating virus entry into the host cells [28]. Global statistical analysis showed significant association of *CLEC4M* variants with COVID positive status, confirming its role in disease susceptibility.

Two potential new risk variants were found in *MAVS* and one in *IFIH1* genes, each in at least two independent families. MAVS is a central adaptor protein which, after interaction with the M protein of SARS-CoV-2, lacks the capability of aggregation and impairs recruitment of downstream targets thus reducing the innate antiviral response [29]. However, our global statistical analysis did not confirm this hypothesis indicating that variants in this gene have minor impact in disease susceptibility unless they add to other risk variants.

Whit respect to *IFIH1* gene, in addition to a potential spice variant, we also detected another missense variant previously reported as a potential susceptibility variant to COVID-19 infection in African and African-American population [30]. IFIH1 primarily regulates IFN induction in response to SARS-CoV-2 infection. Viral intermediates specifically activate the IFN response through IFIH1-mediated sensing [31]. Loss of function of *IFIH1* might impact the viral particles recognition and the production of an appropriate immune response. With respect to the activation of the innate immune response, it is of interest our finding of a rare variant in *NLRP3* (p.Val200Met): although it is reported with conflicting interpretations on ClinVar, a few functional studies suggest that it might upregulate the inflammasome in response to SARS-CoV-2 infection leading to cytokine storm [32].

Other gene variants (in *BIRC2*, *LZTFL1* and *DDX58*) were found in severely affected individuals but did not co-segregate with COVID-positive status. Interestingly, DDX58 mRNA, has been found to be part of the ceRNA (competing endogenous RNA) network, together with other miRNA and long non coding RNAs, which represent one additional machinery that can be exploited by the viral agent upon infection [33]. *LZTFL1* has already been reported in association with severe COVID-19 infection [23], whereas *BIRC2* is an antiapoptotic gene as well as a repressor of the noncanonical NF-κB pathway, and a potent negative regulator of LTR-dependent HIV-1 transcription [34]. Interestingly, also our global statistical analysis showed significant association of *BIRC2* variants with SARS-CoV2 severity: 38.5% of carrier individuals had severe manifestations and 9.1% were asymptomatic.

Our global statistical analysis showed an additional gene, *STAT2*, whose variants are significantly more represented in COVID-positive individuals, beside familial segregation. *STAT2* is a member of the STAT family and plays an essential role in immune responses to extracellular

and intracellular stimuli, including viral invasion: the gene product mediates host defence against viral infections through interferon (IFN)-α/β (IFNα/β) signalling [34].

With respect to protective variants we confirmed previous reports and UCSC records that identified *TMPRSS2* as a protective gene against COVID-19 infection. *TMPRSS2* is expressed in many human tissues and plays a critical role in spreading the infection of viruses including Coronavirus with a central in starting the entire host-pathogen interaction initiated with the physical binding of ACE2 to S-protein [35]. Global statistical analysis could not confirm the association with COVID19 negative status for *TMPRSS2*, likely due to the small sample size.

New potentially protective variants in further genes have been identified in our families, within *IRF7*, *IFNAR1*, *IFITM3* and *NLRP1*. The first three genes are involved in the interferon alpha/beta pathway, whose insufficiency is reported as a severity risk factor in the literature. Interferon is one major player in innate immune response and fine-tuning is crucial to avoid increased or reduced damage upon viral infection. Interestingly, also our global statistical analysis showed significant association of *IFITM3* variants with COVID19 negative status, confirming its role in disease protection. IFITM3 is reported with an ambiguous role, mainly the one of SARS-CoV-2 infection restriction: gain of function variants might increase viral restriction thus protecting the host [36]. *NLRP1* is a direct sensor for RNA virus infection [37] and can modulate the host response. Global statistical analysis could not confirm the association with COVID-19 negative status but it showed a trend for *NLRP1* although it not statistically significant (p = 0.093).

*IRF7* and *IFNAR1* role in host response seems to be ambiguous in our family analysis and in global analysis respectively: for *IRF7* we detected variants in severely affected individuals, suggesting a severity effect, as well as variants in healthy subjects, indicating a protective effect; for *IFNAR1*, familial analysis suggested a protective role, but global statistical analysis indicated an higher frequency of variants in severely affected individuals. It is possible that the effect is variant-specific and depends on up or down regulation of the gene function by that specific variant or that the size of the sample is too small and there are individual additional risk variants.

With respect to rare variants, twelve are quoted as damaging in VarSome and therefore it is likely that they affect the gene function: however, five of them did not segregate with positive or negative status suggesting that their role in host response may not be unique but requires additional variants. The other eight variants were found only in affected subjects although they were not shared by more than two families.

Global analysis showed some variants associated with anosmia in *IFHI1*, *TLR8*, *MAVS a*nd TRAF3: this suggests that not only the severity of symptomatology but also even some very specific clinical manifestations depend on host genomic profiles.

We looked at genes on the X chromosome and did not find any association between the observation of less symptomatic females and variants in *ACE2* gene; along the same line common variants in *TLR7* or *TLR8* where not significantly represented among symptomatic males; however, we found that all carrier males of *TLR7* and *TLR8* variants were symptomatic. This finding suggests a possible role of risk but larger sample size is needed. Indeed, the small size of our sample does not make the statistical data sufficiently robust to highlight some associations due to lack of statistical power.

## Conclusions

Overall, the results of our study confirm that the SARS-CoV-2 phenotype is deeply dependent on host genetic profile, both with a mendelian transmission pattern and a polygenic architecture. In addition, we can also add evidence to the observations of other studies about the

biological mechanisms likely involved in the response to COVID-19 infection, including disrupted autophagy, abnormal activity of receptors for viral antigens or alternative receptors for SARS-CoV-2, alteration in recognition and sensoring of viral particles and activation of the innate immunity, excessive inflammasome activation and cytokine storm, abnormal interaction between virus and the host intracellular RNA network. Many of these nodes are potential targets for precision therapy. Finally, genetic profiling of the host with a COVID-19 gene panel may provide the genetic biomarkers for susceptibility or resistance against the coronavirus infection, which might be useful for identifying the susceptible population groups for targeted interventions and for making relevant public health policy decisions.

## WEB resources

wAnnovar http://wannovar.wglab.org/
   UCSC Genome browser https://genome.ucsc.edu/
   gnomAD https://gnomad.broadinstitute.org/
   Varsome https://varsome.com/
   SpliceAl https://spliceailookup.broadinstitute.org/

## Supporting information

**S1 Table. List of genes in the customized panel.**
(DOCX)

**S2 Table. Familial segregation of all risk variants in families with positive SARS-CoV-2 test.** [a] Human GRCh37/hg19; [b] Minor Allele Frequency. Red: variants already reported as of risk by the COVID-19 HGI in the UCSC Genome Browser.
(DOCX)

**S3 Table. Familial segregation of all protective variants in families with negative SARS-CoV-2 test.** [a] Human GRCh37/hg19; [b] Minor Allele Frequency. Red: variants already reported as of risk by the COVID-19 HGI in the UCSC Genome Browser. Blue: variants already reported as of protection by the COVID-19 HGI in the UCSC Genome Browser.
(DOCX)

**S4 Table. Familial segregation of all variants in families with severe COVID-19.** [a] Human GRCh37/hg19; [b] Minor Allele Frequency. Red: variants already reported as of risk by the COVID-19 HGI in the UCSC Genome Browser.
(DOCX)

**S5 Table. All rare variants (MAF<1%) identified in subjects of our families.** R: Risk variant; S: Severity variant; U: Unclassified; /: Absent. [a] Human GRCh37/hg19; [b] Minor Allele Frequency.
(DOCX)

## Acknowledgments

We thank families for their collaboration for this study.

## Author Contributions

**Conceptualization:** Alessia Azzarà, Fiorella Gurrieri.

**Data curation:** Alessia Azzarà, Roberto Sacco, Fiorella Gurrieri.

**Formal analysis:** Alessia Azzarà, Roberto Sacco.

**Funding acquisition:** Fiorella Gurrieri.

**Investigation:** Alessia Azzarà, Ilaria Cassano, Roberto Sacco.

**Methodology:** Alessia Azzarà, Ilaria Cassano, Carla Lintas, Roberto Sacco.

**Project administration:** Alessia Azzarà, Fiorella Gurrieri.

**Resources:** Alessia Azzarà, Maria Cristina Tirindelli, Carolina Nobile, Valentina Schittone.

**Software:** Alessia Azzarà, Roberto Sacco.

**Supervision:** Alessia Azzarà, Fiorella Gurrieri.

**Validation:** Alessia Azzarà, Roberto Sacco.

**Visualization:** Elisa Paccagnella, Maria Cristina Tirindelli, Carolina Nobile, Valentina Schittone, Carla Lintas, Roberto Sacco.

**Writing – original draft:** Alessia Azzarà, Ilaria Cassano, Carla Lintas, Roberto Sacco, Fiorella Gurrieri.

**Writing – review & editing:** Alessia Azzarà, Fiorella Gurrieri.

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
