## [Decision Letter · Decision Letter 0]

12 Aug 2022

PONE-D-22-07070Genetic variants determine intrafamilial variability of SARS-CoV-2 clinical outcomes in 19 Italian familiesPLOS ONE

Dear Dr. Azzarà,

Thank you for submitting your manuscript to PLOS ONE. After careful consideration, we feel that it has merit but does not fully meet PLOS ONE’s publication criteria as it currently stands. Therefore, we invite you to submit a revised version of the manuscript that addresses the points raised during the review process.

We look forward to receiving your revised manuscript.

Kind regards,

Cinzia Ciccacci

Academic Editor

PLOS ONE

Journal Requirements:

3. In your statement, please include the full name of the IRB or ethics committee who approved or waived your study, as well as whether or not you obtained informed written or verbal consent. If consent was waived for your study, please include this information in your statement as well.

“We thank the Rome Biomedical Campus University Foundation, the ANIA Foundation and the  Fundraising area of the Rome Biomedical Campus University for financial support of this study. We thank families for their collaboration for this study.”

“AA CL received a fellowship for one year by Rome Biomedical Campus University Foundation (AA) and the ANIA Foundation (CL).

https://www.unicampus.it/ateneo/biomedical-university-foundation

https://www.ania.it/

5. Please expand the acronym “ANIA” (as indicated in your financial disclosure) so that it states the name of your funders in full.

8.Please review your reference list to ensure that it is complete and correct. If you have cited papers that have been retracted, please include the rationale for doing so in the manuscript text, or remove these references and replace them with relevant current references. Any changes to the reference list should be mentioned in the rebuttal letter that accompanies your revised manuscript. If you need to cite a retracted article, indicate the article’s retracted status in the References list and also include a citation and full reference for the retraction notice.

Reviewers' comments:

Reviewer's Responses to Questions

**Comments to the Author**

1. Is the manuscript technically sound, and do the data support the conclusions?

Reviewer #1: Yes

2. Has the statistical analysis been performed appropriately and rigorously? 

Reviewer #1: Yes

3. Have the authors made all data underlying the findings in their manuscript fully available?

Reviewer #1: No

4. Is the manuscript presented in an intelligible fashion and written in standard English?

Reviewer #1: Yes

5. Review Comments to the Author

Reviewer #1: Azzará et al. investigated the transmission dynamics of SARS-COV-2 within Italian families whose members responded in different ways to infection, before the availability of any vaccine. To this, they performed a segregation analysis in 19 families of 42 genes involved in immunity and virus enter. They also performed a global statistical analysis. They identified 18 risk variants co-segregating with COVID-positive status and 6 variants with a protective effect. Moreover, 16 variants showed a trend of association to a severe phenotype. The global statistical analysis confirmed positive associations between specific variants and SARS-COV-2 response. The study is interesting, original, and contributes to the search of variants associated with COVID-19 infection and severity. However, they are few concerns with the manuscript in its current form.

- Regarding the global statistical analysis: as the authors acknowledged, the sample size was small for this analysis and some associations could be lost due to lack of statistical power. This needs to be better discussed in the Discussion section as a limitation of the study.

- Page 14, lines 187-188: What did the authors mean by gene mutation rate? Frequency of each variant? If yes, please correct the information in the text since mutation rate is not the same as the frequency of individual variants.

- There are 8 tables (plus 2 figures). Some tables are unnecessary. Table 2 could be added in the Supplementary material and table 7 could be excluded and the information only written in the main text.

- Table 8 (Statistical global analysis of gene variants): This table could be better described and formatted. Are the frequencies shown for the minor alleles? The authors could add OR for each variant. Please check the frequency of the most mutated gene in positive individuals (STAT2). Positive is 0.62% only? And for what variants these frequencies of the most mutated genes in positive and negative individuals are shown?

- Discussion (page 30, line 432): “…whereas it showed a trend for NLRP1 (p=0.093) as a protective gene (data not shown)”. The authors declared that all information was included in the paper; thus, these data not shown must be included in the paper.

- The Discussion section is too long. It could be shortened.

- It would be interesting to cite a recent systematic review and meta-analysis published in Plos One regarding the association of genetic variants and COVID-19:

Genetic polymorphisms associated with susceptibility to COVID-19 disease and severity: A systematic review and meta-analysis. Dieter C et al. PLoS One. 2022 Jul 6;17(7):e0270627. doi: 10.1371/journal.pone.0270627.

6. PLOS authors have the option to publish the peer review history of their article (what does this mean?). If published, this will include your full peer review and any attached files.

Reviewer #1: **Yes: **Daisy Crispim Moreira

---

## [Author Response · Author response to Decision Letter 0]

23 Sep 2022

- Regarding the global statistical analysis: as the authors acknowledged, the sample size was small for this analysis and some associations could be lost due to lack of statistical power. This needs to be better discussed in the Discussion section as a limitation of the study.

A: We included in the text this aspect in the Discussion section (page 27 lines 463-465).

- Page 14, lines 187-188: What did the authors mean by gene mutation rate? Frequency of each variant? If yes, please correct the information in the text since mutation rate is not the same as the frequency of individual variants.

A: We don’t refer to allele frequency, but the "mutation rate" means the number of variants identified in that gene in our population, in other words how much that gene was mutated in different groups of individuals (i.e. gene specific mutation burden). We specified in all Tables the “gnomAD %” as the “gnomAD MAF %”. We have specified this information in the text (page 12 line 192). 

- There are 8 tables (plus 2 figures). Some tables are unnecessary. Table 2 could be added in the Supplementary material and table 7 could be excluded and the information only written in the main text.

A: We agree with the reviewer. We moved Table 2 in the Supplementary Materials (new Suppl. Table 1) and we excluded Table 7. All numbers of the tables were modified: 6 tables (plus 2 figures) in the manuscript and 5 tables in Supplementary tables.

- Table 8 (Statistical global analysis of gene variants): This table could be better described and formatted. Are the frequencies shown for the minor alleles? The authors could add OR for each variant. Please check the frequency of the most mutated gene in positive individuals (STAT2). Positive is 0.62% only? And for what variants these frequencies of the most mutated genes in positive and negative individuals are shown?

A: We agree with the reviewer. We added in the Table 8 (now table 6 in the revised version) the odds ratios with the confidence intervals for each results. We better formatted and described the table in Results of Global analysis section (page 21 lines 312-338). 

Moreover, the percentage refers to individuals carrying that specific variant in the study population, not the MAF (minor allele frequency). The MAF of each the variants were reported in the previously tables of the familial segregation (Tables 2-5). 

STAT2 was the only gene whose variants were detected exclusively in positive individuals (i.e. 2 variants in 2 patients out of mutated).

- Discussion (page 30, line 432): “…whereas it showed a trend for NLRP1 (p=0.093) as a protective gene (data not shown)”. The authors declared that all information was included in the paper; thus, these data not shown must be included in the paper.

A: “Data not shown” refers to the absence of the gene in the table because we included only the data with a significant p-value (0.093 added). We just wanted to highlight the trend of NRLP1 gene in text (moved to page 26 line 443). 

- The Discussion section is too long. It could be shortened.

A: As requested, we shorted the discussion. 

- It would be interesting to cite a recent systematic review and meta-analysis published in Plos One regarding the association of genetic variants and COVID-19:

Genetic polymorphisms associated with susceptibility to COVID-19 disease and severity: A systematic review and meta-analysis. Dieter C et al. PLoS One. 2022 Jul 6;17(7):e0270627. doi: 10.1371/journal.pone.0270627.

A: We have included reference #3

---

## [Editor Report · Decision Letter 1]

27 Sep 2022

Genetic variants determine intrafamilial variability of SARS-CoV-2 clinical outcomes in 19 Italian families

PONE-D-22-07070R1

Dear Dr. Azzarà,

We’re pleased to inform you that your manuscript has been judged scientifically suitable for publication and will be formally accepted for publication once it meets all outstanding technical requirements.

Kind regards,

Cinzia Ciccacci

Academic Editor

PLOS ONE
---

## [Editor Report · Acceptance letter]

4 Oct 2022

PONE-D-22-07070R1 

Genetic variants determine intrafamilial variability of SARS-CoV-2 clinical outcomes in 19 Italian families 

Dear Dr. Azzarà:

I'm pleased to inform you that your manuscript has been deemed suitable for publication in PLOS ONE. Congratulations! Your manuscript is now with our production department. 

Kind regards, 

on behalf of

Dr. Cinzia Ciccacci 

Academic Editor

PLOS ONE